# Associations between gut microbiota and personality traits: insights from a captive common marmoset (*Callithrix jacchus*) colony

Huimin Ye,[1,2,3] Vedrana Šlipogor,[4,5,6] Buck T. Hanson,[1,7] Joana Séneca,[1,8,9] Bela Hausmann,[8,9] Craig W. Herbold,[1,10] Petra Pjevac,[1,8] Thomas Bugnyar,[6] Alexander Loy[1,8]

**ABSTRACT** Recent studies have suggested associations between consistent inter-individual behavioral variation (i.e., animal personality) and gut microbiota. Non-human primates living under controlled conditions are valuable models to investigate diet-independent microbiome-host interactions. In this study, we investigated associations between specific gut microbiota members and personality traits, as well as group membership, sex, age class, breeding status, and relatedness of 26 captive common marmosets (*Callithrix jacchus*), maintained under the same diet and housing conditions. Personality was assessed using an established testing battery in repeated tests. Then, we collected a total of 225 fecal samples during the summers of 2017 and 2019 from five marmoset social groups for 16S rRNA gene amplicon sequencing. Within-individual microbiota variance was smaller than that between group members. Group members also exhibited more similar gut microbiota than individuals from different groups in each sampling year. Beta diversity of the gut microbiota was linked with personality traits, age class, sex, and breeding status, but not with genetic relatedness. We identified specific bacterial taxa associated with personality traits. In particular, members of the sulfite-reducing genus *Desulfovibrio* were enriched in more avoidant marmosets. Amplicon sequencing of the dissimilatory sulfite reductase gene *dsrB* confirmed this pattern, yet additionally revealed an unknown uncultured bacterium that was the predominant sulfite-reducing bacterium in the fecal samples and was linked to more explorative individuals. These findings highlight specific association patterns between identified microbial taxa and personality traits in captive common marmosets.

**IMPORTANCE** This study provides valuable insights into the intricate relationship between gut microbiota and host personality traits, using captive common marmosets as a model. By controlling for diet and housing conditions, it probes key host factors such as personality, age, sex, and social group membership, offering a robust framework for understanding microbiome-host interactions. The discovery of specific microbial taxa associated with personality traits, particularly the enrichment of sulfite-reducing genera in more avoidant individuals, underscores the potential of the gut microbiome to reflect or be associated with personality differences. These findings advance our understanding of microbiome-host dynamics and pave the way for future research on the mechanistic links between behavior and gut microbiota in other animal models and across broader ecological contexts.

**KEYWORDS** gut microbiome, temperament, neotropical primate, sulfite-reducing bacteria

**Peer Reviewer** Aria Hajihassan, Islamic Azad University, Tehran, Tehran, Iran

Address correspondence to Huimin Ye, yehuiminyhm@gmail.com, or Vedrana Šlipogor, vedrana.slipogor@unil.ch.

The authors declare no conflict of interest.

See the funding table on p. 16.

The gut microbiome plays a crucial role in host health by modulating metabolism (1), immunity (2), and behavior and cognition (3). The composition and function of the gut microbiota are influenced by various factors, including host genetics, birth delivery method, geography, age, medication, and diet (4–7). Investigating the potential causal relationships and associations between microbiota variability and these factors has been challenging, particularly in humans and wild non-human animals, due to the difficulty of controlling environmental or dietary variables. Therefore, studying these associations in non-human primate species closely related to humans under controlled conditions in captivity is a useful approach to uncovering the potential drivers of microbial composition.

The common marmoset (*Callithrix jacchus*), a neotropical non-human primate species, has become one of the most commonly studied primates in biomedicine (8), as well as in behavior and cognition research (9, 10). This is due to its genetic proximity to humans, shared physiological and anatomical characteristics, and similar social organization (11). Studies on common marmosets held at different institutions (12, 13) have focused on microbiota variability in captive colonies under different health statuses, social conditions, and laboratory environments (12, 13). These studies have revealed significant variability in the gut microbiota of marmosets, identifying five dominant phyla in various healthy cohorts, namely, Actinobacteria, Bacteroidota, Firmicutes, Fusobacteria, and Proteobacteria (12). Changes in microbial composition have also been documented in many intestinal diseases of common marmosets, such as inflammatory bowel disease (13), chronic diarrhea (14), and strictures (15). However, the specific factors that shape the microbial composition in captive marmosets are not yet comprehensively understood.

The human gut microbiota has been associated with individual variation in cognition, (social) behavior, and personality (16–19). In particular, high gut microbiota diversity has been linked with larger social networks, suggesting a significant role of social interactions in shaping the gut microbiome (18). Additionally, gut microbiota diversity has been associated with positive emotion in *Prevotella*-dominant individuals (20). In captive animals, with uniform diet and environmental conditions, other potential drivers of microbiota variability, such as social behavior or personality traits, can be more precisely investigated. Associations between gut microbiome and social behaviors have been observed in non-human primates. For instance, in infant rhesus macaques (*Macaca mulatta*), breast-fed individuals developed distinct gut microbiomes and immune systems compared to bottle-fed individuals (21). Grooming-based social networks were shown to predict microbiota composition in wild savannah baboons (*Papio cynocephalus*) (22). In free-ranging rhesus macaques, *Faecalibacterium* was positively related to sociability, while less social individuals harbored a higher abundance of potentially pathogenic bacteria species (23).

Recent research has uncovered significant associations between gut microbiota and personality, characterized as consistent inter-individual differences in behavior patterns that remain stable over time and/or across different contexts and/or situations (24, 25), in both humans and non-human animals (16, 18, 19, 26, 27). For instance, high microbiota diversity has been positively linked with exploration, openness, and sociability and negatively with stress and anxiety (16, 18). A recent study on Tibetan macaques (*Macaca thibetana*) reported positive correlations of *Akkermansia*, *Dialister*, and *Asteroleplasma* with sociability scores and links between the gut microbiota and macaque personality (28). Previous research has demonstrated the stability of personality structure in captive marmosets over both short-term (2 weeks) (29, 30) and long-term (4 years) periods (30). Another recent study found that group membership impacted the composition and functional potential of the gut microbiome in wild common marmosets (31). However, to date, no studies have established associations between personality and gut microbiome in marmosets.

Among microbial functional groups, sulfite-reducing bacteria (SRB) are of particular interest because they produce hydrogen sulfide ($H_2S$) during anaerobic respiration (32, 33). $H_2S$ is a biologically active metabolite with complex effects on host physiology: in

rodents, high levels of $H_2S$ have been linked to impaired cognition and stress-related phenotypes, whereas lower levels may exert protective effects (34–36). SRB abundance has also been associated with aging and intestinal disorders in both humans and animals (36, 37). Despite these potential behavioral and physiological roles, the diversity of intestinal SRB is still poorly characterized in non-human primates. We, therefore, included a specific focus on SRB in this study to explore whether variation in this functional group is linked to personality traits in captive common marmosets.

In this study, we aimed to identify the factors driving gut microbiota variability in captive common marmosets. We analyzed 225 fecal samples from 26 individuals living in five social groups, all maintained under the same controlled diet and housing conditions over 2 years. We linked gut microbiota variability, analyzed by 16S rRNA gene amplicon sequencing, to individual characteristics, like group membership, age class, sex, relatedness, and breeding status, and previously obtained personality profiles (38), assessed by five repeated personality tests. We hypothesized that (i) within-individual microbiota variance would be smaller than between-individual variance; (ii) group members would share a more similar gut microbiome composition than non-group members; (iii) age, sex, relatedness, and breeding status would impact the microbiome composition; and (iv) personality traits, including Exploration-Avoidance, Boldness-Shyness, and Stress/Activity, would be associated with gut microbiota. Additionally, we investigated the diversity of SRB by amplicon sequencing of their marker gene *dsrB*, encoding dissimilatory sulfite reductase.

## MATERIALS AND METHODS

### Animals

Twenty-six common marmosets born in captivity and housed in five different social groups at the Department of Behavioral and Cognitive Biology, Faculty of Life Sciences, University of Vienna, Austria, participated in this study. The marmosets were classified into three age classes: juveniles (age < 1 year), adults (12 years > age ≥ 1 year), and older adults (hereafter labeled "old"; age ≥ 12 years). This division was made as marmosets in the wild rarely live longer than 12 years (39). Each social group lived in an indoor cage (250 × 250 × 250 cm) connected to an outdoor cage (250 × 250 × 250 cm). These wire mesh cages were connected to an experimental compartment via a passageway system of tunnels with movable doors. Wood pellets were used as floor bedding in each indoor cage, and different climbing and resting structures and objects (i.e., branches, ropes, blankets, textiles, sleeping baskets, and boxes) were placed in the cages. Room temperature was maintained at 21°C–29°C, and humidity was controlled at 30%–60%. A 12:12 h light:dark cycle was maintained throughout the study. During personality testing, the animals were fed daily in the morning hours with New World primate biscuits (Mazuri food, USA) and at noon with a selection of seasonal fruits, vegetables, grains, milk products, marmoset jelly, marmoset gum, protein, vitamin supplements, and insects (mealworms and crickets). During the fecal sample collection, the animals were fed immediately after taking the samples in the morning (i.e., around 11 a.m.), with food they usually received for breakfast and lunch. This diet was consistently maintained throughout the sampling period. All common marmosets had *ad libitum* access to water.

### Fecal sample collection and DNA extraction

We analyzed a total of 225 fecal samples collected during two periods: May to August in 2017 and June to August in 2019. Marmosets were released into tunnels, where movable doors were used to separate each individual into different sections of the tunnel. Feces fell through the tunnel onto 70% ethanol-sterilized trays positioned below. Each individual was kept in the tunnel for 15 min, with visual access to the group members to alleviate any possible stress, and feces were collected immediately after defecation. Subsamples were preserved in 1.5 mL microcentrifuge tubes and stored at

−20°C (Table S1). Nucleic acids were extracted from feces as previously described (40). Briefly, thawed samples were resuspended in 0.5 mL cetyltrimethylammonium bromide buffer (pH 8) and transferred to a Lysing Matrix E tube (MP Biomedicals, USA) containing 0.5 mL phenol:chloroform:isoamyl alcohol (25:24:1; pH 8, Carl Roth GmbH, Germany). Cells were lysed by bead-beating (FastPrep-24 bead beater, MP Biomedicals, Heidelberg, Germany) for 30 s and cooled on dry ice, followed by centrifugation for 5 min at 4°C (16,000 × $g$). Aqueous supernatants were transferred to a sterile 1.5 mL microcentrifuge tube containing 0.5 mL chloroform:isoamyl alcohol (24:1, Carl Roth GmbH, Germany) and mixed. Samples were centrifuged for 5 min at room temperature (16,000 × $g$), and aqueous supernatants were removed to a sterile 1.5 mL microcentrifuge tube. One milliliter polyethylene glycol/NaCl solution (30% [wt/vol] polyethylene glycol 8000, 1.6 M NaCl) was added to precipitate nucleic acids at room temperature for 2 h. Nucleic acids were precipitated by centrifugation for 10 min at 4°C (18,000 × $g$) and washed twice with ice-cold 70% molecular-grade ethanol (Sigma-Aldrich, St. Louis, MO, USA). Nucleic acid pellets were air-dried under ambient room conditions and resuspended in 50 µL Tris-EDTA (pH 8.0) buffer and stored at −20°C for subsequent 16S rRNA gene amplicon sequencing. A subset of DNA samples collected in 2017 ($n = 75$) was selected for *dsrB* gene amplicon sequencing, ensuring that most individuals were included at least once.

## 16S rRNA gene and *dsrB* amplicon sequencing

Amplicon sequencing of the 16S rRNA gene hypervariable V3–V4 regions was performed after amplification and barcoding with primers 341F and 785R (41). Sequencing of the beta subunit of dissimilatory sulfite reductase (*dsrB*) gene was carried out with primers DSR1762Fmix and DSR2107Rmix (42). A two-step amplification and barcoding approach was employed for samples collected in 2017 and Pjevac et al. (43) for samples collected in 2019. Barcoded libraries were pooled at equivalent copy numbers ($20 \times 10^9$) and paired-end sequenced on an Illumina MiSeq (V3 chemistry, 600 cycles). Raw data processing was performed as described previously (43). Amplicon sequence variants (ASVs) were inferred using the DADA2 R package version 1.42, applying the recommended workflow (44). FASTQ reads 1 and 2 were trimmed at 230 nt with allowed expected errors of 4 and 6, respectively. ASV sequences were subsequently classified using DADA2 and the SILVA database SSU Ref NR 99 release 138.1 (44, 45) using a confidence threshold of 0.5.

The *dsrB* gene sequencing data were analyzed according to the procedures outlined previously (42) and clustered into operational taxonomic units (OTUs) based on a nucleotide sequence similarity cutoff of 97% using Uparse (46). Representative *dsrB* sequences were classified using phylogenetic placement with a curated *DsrB* gene reference sequence database and corresponding consensus tree (42, 47).

## Microbiome profiling

Sequencing data were analyzed using the software packages Phyloseq (version 1.48.0) (48) and microeco (version 1.7.1) (49) in R (version 4.4.0). For alpha-diversity (Shannon and Simpson indices) and beta-diversity (Bray-Curtis distances) analyses, the OTU/ASV tables were rarefied at the depth of the smallest library size. Differential analysis of taxa was performed using the random forest method (50, 51), a machine learning program that can identify an optimal set of taxa with high discriminative power.

## Linear mixed models and Mantel test to identify factors that shape gut microbiota composition in common marmosets

We computed linear mixed models (LMMs) using the function lmer of the lme4 package (version 1.1.35.3) with the optimizer "bobyqa" (52). For all models, covariates were z-transformed to improve model fit. Random slopes were incorporated to maintain Type I error rates at the nominal level of 5% (53). After fitting each linear mixed-effects regression (lmer) model, we verified the assumptions of normality, homoscedasticity,

and collinearity and assessed model stability. *P*-values for individual effects were derived from likelihood ratio tests comparing the full model to the respective null or reduced models (using the ANOVA function with the "Chisq" test argument), facilitated by the drop1 function (53). Null models included only intercepts, random effects, and random slopes, while reduced models also incorporated assigned control factors. We calculated effect sizes for the full models, considering both fixed and random effects, using the r.squaredGLMM function from the MuMIn package (version 1.43.17) (54). Confidence intervals were estimated through parametric bootstrapping with an adjusted bootMer function from the lme4 package. All models used in this study are listed in Table S2.

### Data preparation

The 16S rRNA gene ASV table produced by the DADA2 pipeline was normalized using geometric means of pairwise ratios (GMPR; 0.1.3) (55), a method designed for zero-inflated count data, and was applied to microbiome sequencing data. A phylogenetic tree was built using representative ASV sequences. ASV sequences were aligned using the package DECIPHER (version 3.0.0) (56), and the tree was built with the package phangorn (version 2.22.1) (57). A midpoint rooted tree was used to calculate the alpha diversity index phylogenetic diversities with the picante package (version 1.8.2) (58). The generalized UniFrac (GUniFrac) distances were calculated for beta diversity using the GUniFrac package (version 1.8) (59). All statistical analyses were conducted in R (version 4.4.0).

### Mantel test—beta diversity and group membership

We conducted Mantel tests (60) to investigate the relationship between group membership and beta diversity using 1,000 permutations. Mantel tests were performed on samples from each sampling year, yielding the mean absolute differences in dissimilarities within and between the groups.

### LMM I—beta diversity within the same individual

Only dyads of individuals within the same social group in the same year were selected for analysis to investigate if the same individual has more similar gut microbiota compared to other group members. The mean value of GUniFrac distances of the same individual in the same year was used as a response; the factor "Same ID" was used as a fixed factor, "IDDyad (e.g., monkey A-B)" and "Group ID" were used as random effects. The "sampling year" was used as a control factor and as a random slope.

### LMM II—beta diversity within related individuals

We investigated the effect of maternal relatedness on gut microbiome similarity. All the genetic histories of the individuals were well documented. We assigned relatedness coefficients (RCs) to different kinships: RC = 0.8 for twins, RC = 0.5 for mother-offspring and siblings, and RC = 0.25 for half-siblings (Fig. 1). The mean GUniFrac distances of individual dyads per year were used as a response, the relatedness coefficient between these individuals was used as a fixed effect, and individual dyad (e.g., monkey A-monkey B) was used as a random effect.

### LMM III—beta diversity within individuals with the same breeding status

We investigated the effect of breeding status on gut microbiota similarity. Marmosets were assigned to breeder and helper groups based on their breeding status within the social group. The mean GUniFrac distances of gut microbiota between group members per sampling year were used as a response. Breeding status (BreedingDyad, e.g., breeder-helper) was used as a fixed effect, and "GroupDyad (e.g., group A-group B)" and "IDDyad" were used as random effects.

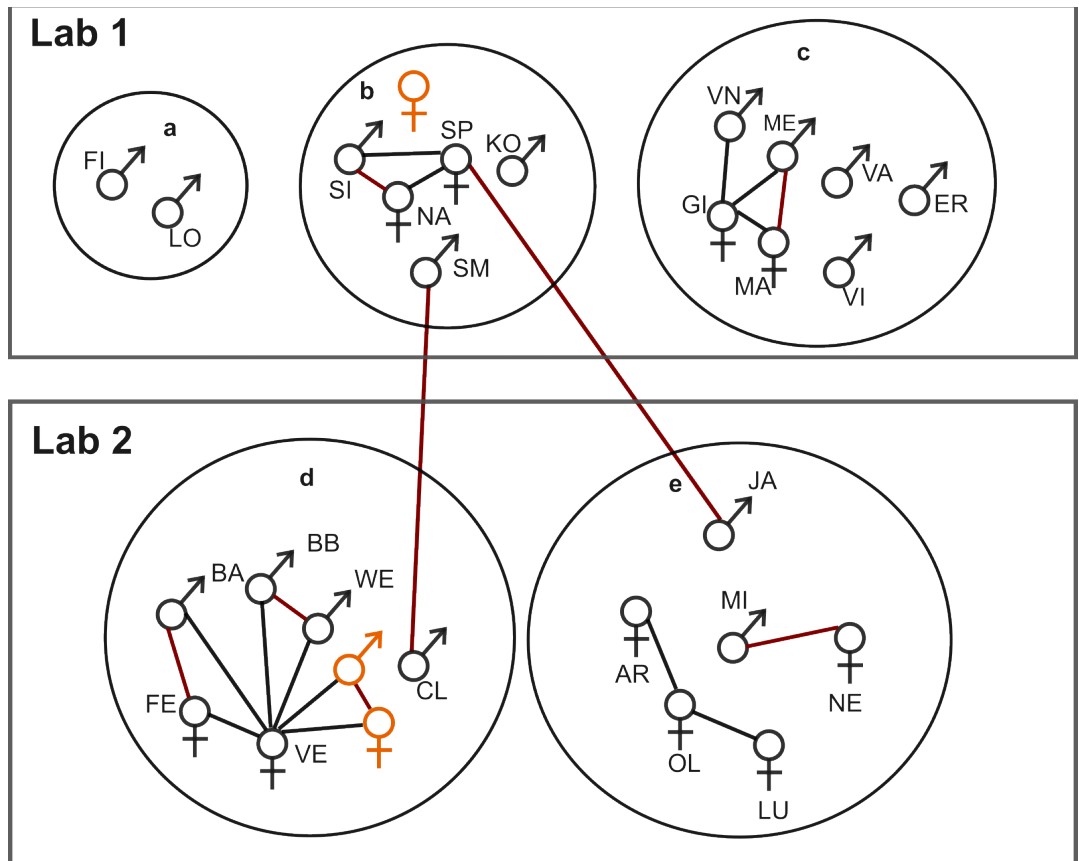

**FIG 1** Social group structure and genetic relatedness of the animals. All animals are allocated to two adjacent lab rooms. The animals are housed in social groups (a–e) in separate cages, ensuring they do not have visual contact with other family groups. Sex symbols represent individual animals, each identified by a name abbreviation. The relationship between individuals is indicated by connecting lines. Red lines: twins; black lines: mother and children. Two newborn animals in group D and a newborn animal in group B after the first sampling period (2017 summer) are marked in orange. After the 2017 summer, KO moved from group B to group E, and GI from group C passed away.

## LMM IV—effect of sampling year, age, and sex on beta diversity

We examined the potential correlation of microbiota similarity between group members, age classes, sexes, and sampling years. The mean GUniFrac distances between group members per year were used as a response, and age class (e.g., juvenile-adult), sex dyads (e.g., female-female), and sampling year (e.g., 2017–2019) were used as predictors. Relatedness was included as a control factor. Individual dyads and group ID were used as random effects.

### Influence of personality traits on gut microbiota

Personality tests of the same marmoset colony were performed in 2016, and the results were published for 27 marmosets (38). To briefly summarize, behavioral variables across five personality tests, namely, Activity, Novel Object, Novel Food, Predator, and Foraging Under Risk, fell within the higher repeatability range of personality studies. The personality characterization obtained by the principal component analysis (PCA) resulted in three personality components: "Exploration-Avoidance" (36.8% of variance; approach and manipulation of novel stimuli), "Boldness-Shyness" (19.9% of variance; responses to risk and predator cues; proximity and visual attention to stimulus), and "Stress/Activity" (14.2% of variance; locomotor activity and stress-related behaviors) (Table S3). In this study, individuals were divided into groups based on their personality scores on principal components (e.g., Exploration and Avoidance). Thus, individuals with positive

values of personality scores were assigned to Exploration, Shyness, and Higher Stress/Activity groups, and those with negative values of personality scores were classified into Avoidance, Boldness, and Lower Stress/Activity groups. Random forest was applied in an exploratory manner to rank taxa by MeanDecreaseGini. Taxa with MeanDecreaseGini > 0 were retained as differential taxa between groups.

Microbiome Multivariable Association with Linear Models (Maaslin2) (61) was employed to test the associations between specific personality trait scores and microbial abundance at the genus level. In this analysis, we used the exact personality scores for correlation analysis rather than categorizing them into positive or negative groups. To account for confounding effects, sex, age class, breeding status, and social group membership were included as fixed effects in the model in addition to the personality traits. Significance was assigned to relationships with a false discovery rate < 0.25.

### *dsrB* phylogenetic analyses

Representative sequences from each OTU were used for phylogenetic analyses. The reference *dsrB* gene sequences were obtained from a *dsrB* gene database (62). Collected *dsrB* gene sequences were aligned with MAFFT (version 7.520) (63) and trimmed using TrimAl (version 1.4) (64) with the flag "-automated 1." Maximum-likelihood trees were created using the IQ-TREE web server with automatic substitution model selection and ultrafast bootstrapping (1,000×) (65–67). The trees were visualized and annotated with iTOL (version 6.9) (68). We examined the relationship between the beta diversity of SRB and sulfate concentrations in the fecal samples ($n$ = 68) using the Mantel test with permutations = 9,999.

### Fecal short-chain fatty acid analysis

Fecal samples collected in 2017 ($n$ = 123) were used for short-chain fatty acid (SCFA) measurement. Fecal samples were diluted to the same concentration with sterile water (0.05 g feces/µL) and centrifuged for 10 min at 4°C (21,000 × $g$). Supernatants were collected into clean 1.5 mL microcentrifuge tubes for SCFA measurement. SCFAs were measured with a capillary electrophoresis P/ACE–MDQ (Beckman Coulter, Krefeld, Germany) equipped with a UV detector with a 230 nm wavelength filter. The capillary was a fused silica column (TSP075375; 75 µm ID, Polymicro Technologies), 60 cm long (50 cm to the detector) and 75 µm in diameter. Samples were prepared for analysis using a CEofix Anions 5 Kit (Beckmann Coulter, Krefeld, Germany) according to the manufacturer's instructions. Analytes were separated in reverse polarity mode at 30 kV (ramp: 0.5 kV/s) for 10 min. All samples were diluted 1:10 with a working solution consisting of 0.01 M NaOH, 0.5 mM $CaCl_2$, and 0.1 mM caproate (internal standard). An external standard mixture of sodium sulfate and the SCFAs, formate, succinate, acetate, lactate, propionate, butyrate, and valerate (1 mmol each), was used for quantification.

## RESULTS

### Gut microbiota composition of common marmosets

Over a period of 2 years, we collected 225 fecal samples from five groups of captive common marmosets and analyzed their gut microbiota composition using 16S rRNA and *dsrB* gene amplicon sequencing. The marmoset social group structure and their genetic relationships are depicted in Fig. 1, and the sample demographics are listed in Table S1. The marmoset fecal microbiome was dominated by Bacteroidota (mean ± S.D. = 56% ± 10.6%), followed by Firmicutes (21.7% ± 7.7%), Actinobacteria (9.9% ± 5.5%), and Proteobacteria (6.8% ± 5.3%) (Fig. S1a). At the family level, Bacteroidota was predominantly composed of *Bacteroidaceae*, *Prevotellaceae*, and *Tannerellaceae*. Firmicutes were mainly represented by *Selenomonadaceae* and *Acidaminococcaceae*. Actinobacteriota primarily consisted of *Bifidobacteriaceae*, while Proteobacteria was mainly represented by *Sutterellaceae* and *Enterobacteriaceae* (Fig. S1b). At the genus level, *Bacteroides*,

*Prevotella_9*, *Parabacteroides*, *Megamonas*, *Bifidobacterium*, and *Phascolarctobacterium* collectively contributed to more than 50% of the total microbial abundance (Fig. S1c).

Variations in microbial composition were primarily observed across different social and age groups, with less variation noted in different sex and breeding status groups (Fig. 2; Fig. S2). Alpha diversity was not significantly affected by social group, age, sex, or breeding status (Fig. 2; Fig. S2).

## Beta diversity of gut microbiota is associated with individual identity, social group, age, sex, and breeding status, but not with genetic relatedness

Linear mixed model analysis revealed that there was a significant effect of individual identity on the beta diversity of gut microbiota in the captive common marmoset colony (LMM I; likelihood ratio test comparing full and reduced model $\chi^2$ = 26.53, df = 1, $P$ < 0.001, $R^2_{m/c}$ = 0.44/0.65) (Table S4). Microbiota composition of fecal samples from the same individual (GUniFrac distance = 0.37 ± 0.06, mean ± SD) was more similar than that of samples from other individuals (GUniFrac distance = 0.42 ± 0.06, mean ± SD) of the same social group. The microbiota composition was also more similar within the same social group than between different social groups in both 2017 (Mantel test: $N_{samples}$ = 123, $N_{individuals}$ = 26, $\bar{x}_{same\ group}$ = 0.371, $\bar{x}_{different\ group}$ = 0.386, $P$ = 0.019) and 2019 (Mantel test: $N_{samples}$ = 105, $N_{individuals}$ = 28, $\bar{x}_{same\ group}$ = 0.445, $\bar{x}_{different\ group}$ = 0.471, $P$ = 0.002) (Table S5). In contrast, relatedness was not linked with the microbiota similarity in common marmosets (LMM II, Table S6), suggesting that maternal relatives, twins, or siblings did not share a more similar microbiota than unrelated individuals. The model examining correlations of GUniFrac distances of microbial composition with breeding status was significant (LMM III; likelihood ratio test comparing full and reduced model $\chi^2$ = 9.008, df = 1, $P$ = 0.003, $R^2_{m/c}$ = 0.37/0.74, Table S7). In particular, individuals with the same breeding status (e.g., both breeders or both helpers) had more similar gut microbiota compared to individuals with different breeding statuses (Fig. S3a). Furthermore, the model examining correlations of dyadic GUniFrac dissimilarity with sampling year, sex, and age class was likewise significant (LMM IV; likelihood ratio test comparing full and reduced model $\chi^2$ = 33.163, df = 7, $P$ < 0.05, $R^2_{m/c}$ = 0.48/0.77, Table S8). Juvenile marmosets exhibited a more similar gut microbiota than other age group dyads, while high divergence of gut microbiota was observed in the older marmoset group (Fig. S3b). Finally, male marmoset dyads had a higher similarity in gut microbiota than other female-female or female-male dyads (Fig. S3c).

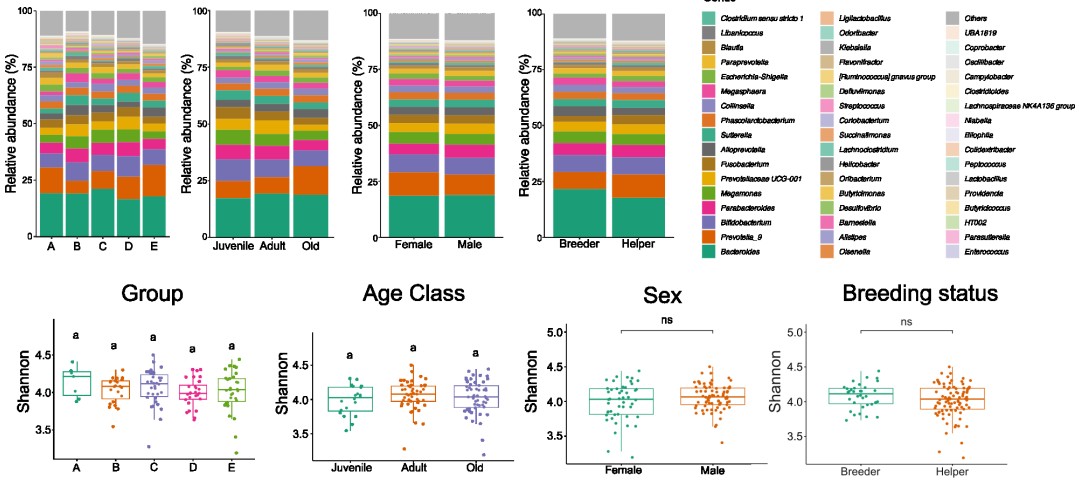

**FIG 2** Microbial variation between social groups, age classes, sex, and breeding status groups in 2017. Averaged relative abundances at the genus level were grouped by social group, age class, sex, and breeding status (upper panel). No significant differences were observed in alpha diversity between different social, age class, sex, and breeding groups (lower panel). Shannon's diversity was compared using Kruskal-Wallis one-way ANOVA with Dunn's multiple comparison test or *t*-test.

Next, we performed a differential analysis based on the random forest algorithm to pinpoint microbiota differences at the genus level across different monkey groups. Different genera were identified as predictors for the same social groups in 2017 and 2019, which might have been due to group structure changes (i.e., as there were newborns in the group and one elder marmoset died between the sampling years [Fig. S4a and b]. Yet, *Helicobacter* and *Desulfovibrio* were consistently identified in both 2017 and 2019 as predictor genera for juvenile and older adult marmosets, respectively. Moreover, the relative abundances of *Succinatimonas*, *Prevotellaceae* UCG-001, and *Parabacteroides* were higher in juvenile and adult marmosets than in older adult marmosets (Fig. S4c and d). No consistent predictor genus was identified for female and male marmosets in 2017 or 2019 (Fig. S4e and f). Additionally, *Bacteroides*, *Bilophila*, *Erysipelatoclostridiaceae* UCG-004, and *Eubacterium* were enriched in breeders in both 2017 and 2019. However, no consistent predictor genus was identified in helpers across the 2 years (Fig. S4g and h).

## Microbial variance between different personality groups

We next investigated the association between personality traits and the gut microbiota. Individuals were divided into groups based on their personality scores on principal components. Thus, individuals with positive personality scores were assigned to Exploration, Shyness, and Higher Stress/Activity groups; those with negative personality scores were classified into Avoidance, Boldness, and Lower Stress/Activity groups. Exploration-Avoidance, Boldness-Shyness, and Stress/Activity personality traits had no impact on alpha diversity. However, we observed significant differences in beta diversity in all three personality traits (Fig. 3a, c, and e). Analysis of the Bray-Curtis distance between the Boldness and Shyness groups showed no significant difference. Thus, the Jaccard distance is shown instead (Fig. 3c). Differential analysis based on the random forest algorithm revealed indicator taxa at the genus level among different personality groups (Fig. 3b, d, and f). The mean decrease Gini coefficient is a measure of how each taxon contributes to the classification of each group (69). The higher the mean decrease in Gini scores, the more important the variable in the model. In the marmosets scoring high on Exploration, four genera, including *Megasphaera*, *Prevotella_9*, *Libanicoccus*, and *Bifidobacterium,* were prominently enriched (Mean Decrease Gini > 2). Conversely, *Bacteroides*, *Desulfovibrio*, *Akkermansia*, *Alistipes*, and *Paenirhodobacter* were enriched in the avoidant group of marmosets (Fig. 3b). The marmosets scoring high in Shyness had an enrichment of seven taxa (*Helicobacter*, *Fusobacterium*, *Butyricicoccus*, *Novosphingobium*, *Clostridioides*, *Paenirhodobacter*, and *Holdemania*), while the bolder marmosets had only *Rubrivivax* enriched (Fig. 3d). In the marmosets scoring high in Stress/Activity, four genera, including *Parabacteroides*, *Prevotellaceae* UCG-001, *Sutterella*, and *Catenisphaera,* were enriched. The gut microbiota of marmosets that scored lower on this trait were characterized by the enrichment of *Butyricimonas*, *Bilophila*, *Oscillibacter*, and *Eisenbergiella* (Fig. 3f).

Next, we performed a Maaslin2 analysis to identify specific microbial taxa significantly associated with personality traits, considering confounding factors such as social group, age class, sex, and breeding status. Maaslin2 analysis revealed 28 taxa significantly associated with three aforementioned personality traits. Among them, seven taxa exhibited a significant negative correlation with Exploration-Avoidance PCA scores, including *Bacteroides* ($r = -0.24$, $q = 0.01$, $P < 0.001$), *Paenirhodobacter* ($r = -0.40$, $q = 0.11$, $P = 0.002$), *Parasutterella* ($r = -0.29$, $q = 0.16$, $P = 0.004$), *Anaerotruncus* ($r = -0.31$, $q = 0.19$, $P = 0.008$), *Staphylococcus* ($r = -0.25$, $q = 0.19$, $P = 0.008$), *Niabella* ($r = -0.33$, $q = 0.21$, $P = 0.01$), and *Helicobacter* ($r = -0.43$, $q = 0.23$, $P = 0.01$). In contrast, *Prevotella_9* ($r = 0.44$, $q = 0.09$, $P = 0.002$) and unclassified *Bacteroidales* ($r = 0.50$, $q = 0.16$, $P = 0.004$) were significantly positively correlated with the Exploration-Avoidance scores (Table S9; Fig. S5). Unclassified *Muribaculaceae* ($r = -0.61$, $q = 0.01$, $P < 0.001$), unclassified *Barnesiellaceae* ($r = -0.85$, $q = 0.09$, $P = 0.001$), unclassified *Prevotellaceae* ($r = -0.68$, $q = 0.15$, $P = 0.004$), unclassified *Tannerellaceae* ($r = -0.56$, $q = 0.15$, $P = 0.004$), and *Pseudorhodoferax* ($r$

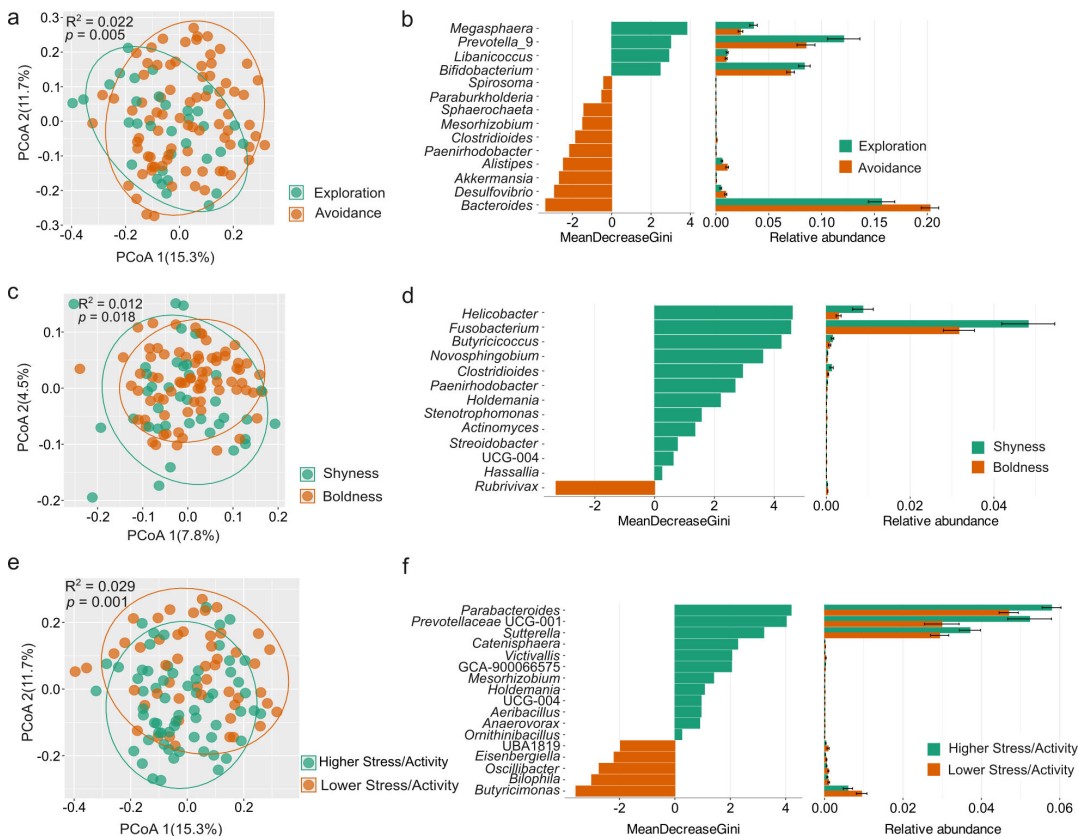

**FIG 3** Variation of beta diversity and the differential taxa among different personality groups. Beta-diversity analysis based on Bray-Curtis distance (a and e) and Jaccard distance (c). Differential taxa between different personality groups (b, d, and f). Mean Decrease Gini measures the extent to which the identified taxa contribute to the classification of personality groups.

= −0.57, $q = 0.18$, $P = 0.006$) were significantly negatively associated, whereas *Helicobacter* ($r = 0.81$, $q = 0.12$, $P = 0.002$), *Butyricicoccus* ($r = 0.68$, $q = 0.23$, $P = 0.01$), and *Martelella* ($r = 0.37$, $q = 0.23$, $P = 0.01$) were significantly positively associated with Boldness-Shyness scores. Ten taxa, including *Victivallis* ($r = −0.88$, $q = 0.02$, $P < 0.001$), UBA1819 ($r = −1.14$, $q = 0.03$, $P < 0.001$), *Eisenbergiella* ($r = −1.06$, $q = 0.05$, $P < 0.001$), *Anaerotruncus* ($r = −0.72$, $q = 0.08$, $P = 0.001$), *Paraburkholderia* ($r = −0.48$, $q = 0.09$, $P = 0.002$), unclassified *Muribaculaceae* ($r = −0.54$, $q = 0.13$, $P = 0.003$), *Paenirhodobacter* ($r = −0.70$, $q = 0.16$, $P = 0.004$), *Alistipes* ($r = −0.64$, $q = 0.19$, $P = 0.008$), *Paracoccus* ($r = −0.75$, $q = 0.19$, $P = 0.006$), and *Rhizobium* ($r = −0.58$, $q = 0.20$, $P = 0.009$) were significantly negatively correlated, while unclassified *Bacteroidales* ($r = 85$, $q = 0.21$, $P = 0.01$) was positively correlated with Stress/Activity scores.

## The composition of sulfite-reducing bacteria is associated with the Exploration-Avoidance personality trait

Amplicon sequencing of the dissimilatory sulfite reductase *dsrB* gene, encoding the beta-subunit of a key microbial enzyme in the production of hydrogen sulfide ($H_2S$), was used to investigate the composition of SRB in the feces of common marmosets. Four operational taxonomic units (i.e., OTU 1, OTU 2, OTU 3, and OTU 4) dominated the *dsrB* diversity, collectively constituting approximately 98% of the sequencing reads across all samples (Fig. 4a). OTU 1 was identified as a yet uncultured bacterium, with its closest relative *Peptococcus niger* (81.5% *dsrB* similarity), a species commonly found in the human gut microbiota (70) (Fig. 4b). OTU 2, OTU 3, and OTU 4 were classified as members of the *Desulfovibrionaceae* family. Based on the proposed 90% *dsrB* sequence identity cutoff for species (42), OTU 2 and OTU 4 were identified as *Desulfovibrio* species

(91.8% and 90.3% *dsrB* similarity, respectively), and OTU 3 was identified as *Bilophila wadsworthia* (97.0% *dsrB* similarity) (Fig. 4b).

We observed no effect of social group, age class, sex, and breeding status on the beta diversity of SRB in marmosets. We then investigated the effect of personality traits on the diversity and composition of the SRB community in the common marmosets. Only Exploration-Avoidance showed a significant impact on the alpha diversity of SRB with a lower Shannon index ($P = 0.003$) and Simpson index ($P = 0.003$) in the more explorative group compared to the more avoidant group (Fig. 5a and b). SRB beta diversity showed significant differences between individuals of the different Exploration-Avoidance groups (Fig. 5c). Differential analysis based on the random forest algorithm was employed to determine differences in the relative abundance of individual *dsrB* OTUs across different groups (Fig. 5d). SRB diversity in the Exploration group was enriched in OTU 1. The Avoidance group was characterized by the enrichment of OTUs belonging to *Desulfovibrionaceae* (OTU 2, 3, 4, and 15) and *Desulfobulbaceae* (OTU 102). The enrichment of *Desulfovibrio* (OTU 3) in the Avoidance group compared to the Exploration group was consistent with the results of the 16S rRNA gene sequence analysis (Fig. 3b). Additionally, the Mantel test revealed a positive correlation between the beta diversity of SRB and fecal sulfate concentrations (Spearman, $r = 0.11$, $P = 0.02$).

## Correlation between gut microbiota, SCFAs, and sulfate

As the mechanism by which gut microbiota affect the host's personality could be, in part, mediated by short-chain fatty acids, we determined the association between individual fecal SCFAs and microbiota members (Fig. 6). The relative abundance of 24 of the 50 most abundant genera was correlated with the concentration of at least one SCFA (Fig. 6). *Megasphaera* ($P = 0.01$) and *Bifidobacterium* ($P = 0.04$), which were enriched in the more explorative marmosets, were positively correlated with fecal sulfate concentration. In contrast, taxa enriched in the more avoidant marmoset group, including *Alistipes* ($P < 0.001$) and *Desulfovibrio* ($P < 0.001$), were negatively correlated with sulfate concentration. *Fusobacterium*, identified as a predictor taxon in shyer marmosets, was negatively correlated with formate ($P = 0.01$) and valerate ($P = 0.04$) and positively correlated with sulfate ($P = 0.045$). Two predictor taxa in the more stressed/active marmoset group exhibited distinct associations with SCFA concentrations. *Prevotellaceae* UCG-001 was positively correlated with formate ($P = 0.02$), propionate ($P < 0.001$), and sulfate ($P = 0.01$). In contrast, *Sutterella* was negatively correlated with butyrate ($P = 0.048$), lactate ($P = 0.049$), and valerate ($P < 0.001$). *Butyricimonas*, a predictor taxon for the lower stress group, was negatively correlated with sulfate ($P < 0.001$) and positively correlated with valerate ($P = 0.006$).

## DISCUSSION

In the present study, we explored the effects of personality traits, as well as social group, relatedness, sex, age class, and breeding status, on gut microbiome composition and variance in captive common marmosets (*Callithrix jacchus*) over 2 years. Our results revealed that within-individual microbiome variance was smaller than between-individual variance and that group members had more similar gut microbiota than non-group members. The personality of our study subjects, as well as their age class, sex, and breeding status, was linked with the beta diversity of the gut microbiota, while alpha diversity was unassociated with these factors. Relatedness neither impacted alpha nor beta diversity in our study colony. Notably, our analysis identified an uncultured bacterium (*dsrB*-OTU 1) as the dominant sulfite-reducing bacterium in this population of marmosets. This study provides the first glimpse into the SRB diversity and the potential effects of various intrinsic and extrinsic factors on the gut microbiota in captive common marmosets under controlled diet and housing conditions.

Previous studies in healthy marmosets have demonstrated significant plasticity in gut microbiota across institutions, with dominance observed in one of five phyla: Actinobacteria, Bacteroidota, Firmicutes, Fusobacteria, or Proteobacteria (12). The gut microbiota of

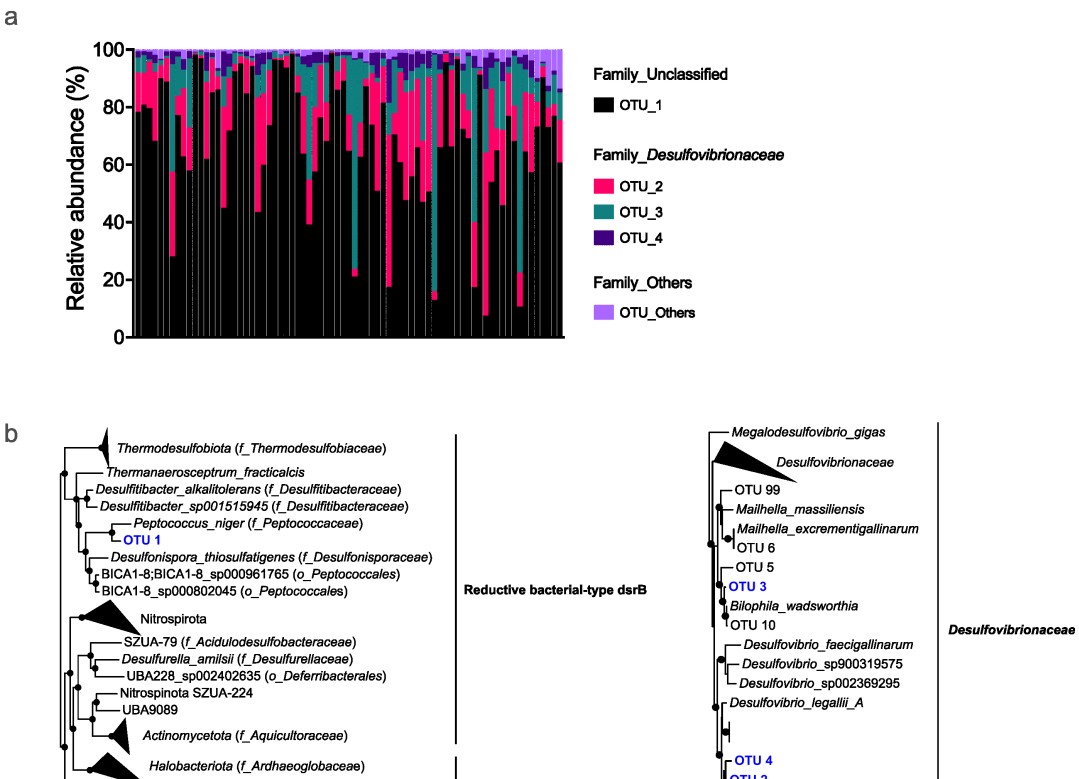

**FIG 4** Composition and phylogeny of sulfite-reducing bacteria. (a) Relative abundance of SRB in the common marmoset colony. Each column shows a sample (*n* = 68). (b) Maximum likelihood trees based on *dsrB* gene sequences (nucleic acid) obtained by *dsrB* gene amplicon sequencing. The phylogenetic analyses were carried out by using one representative sequence from each OTU. The reference *dsrB* sequences were obtained from Diao et al. (62). The tree is midpoint rooted. The four most abundant OTU IDs from this study are in blue. Bootstrap values > 90% are presented on nodes as black-filled circles. The scale bar represents 30% (left) and 20% (right) sequence divergence, respectively.

our marmoset colony was dominated by one of these phyla across the 2-year study period, namely, Bacteroidota, recognized for its ability to break down complex carbohydrates. This is consistent with the diet provided to our colony, which includes tree exudates (i.e., gum) and seasonal fruits rich in complex plant polysaccharides.

Sulfite-reducing bacteria, such as *Desulfovibrio* and *Bilophila* species, were detected in our marmoset colony using 16S rRNA and *dsrB* gene amplicon sequencing. *Desulfovibrio*, *Desulfobacter*, and *Bilophila* were previously identified as SRB in humans and other animals (71, 72). A previous study that investigated SRB composition in sooty mangabeys (*Cercocebus atys*) and baboons (*Papio hamadryas*) has shown that SRB composition is host species-specific in these primates (73). Additionally, *Desulfovibrionales* have been identified as the main SRB in black howler monkeys (*Alouatta caraya*) (74). In our study colony, *dsrB* OTU 1, which shares 81.5% similarity with *Peptococcus niger*, was identified as the dominant SRB. Another *Peptococcus* species, *Peptococcus simiae*, has been isolated from rhesus macaque feces and is known to produce hydrogen sulfide (75). Both *P. simiae* and *P. niger* can produce $H_2S$ from taurine or sulfite. Taurine is an important source of sulfite for SRB in the gut, derived from the diet, but primarily comes from microbial deconjugation of taurine-conjugated bile acids (76). In fact, plasma taurine concentrations in marmosets are significantly higher compared to humans (77). *Bilophila wadsworthia*, represented by *dsrB* OTU 3, is a ubiquitous taurine-respiring bacterium in the human gut (33). We thus hypothesize that the abundant SRB detected in our study use taurine as their main electron acceptor source for anaerobic respiration. Although microbial sulfate and short-chain fatty acids were measured, and *Desulfovibrio* was found to be negatively

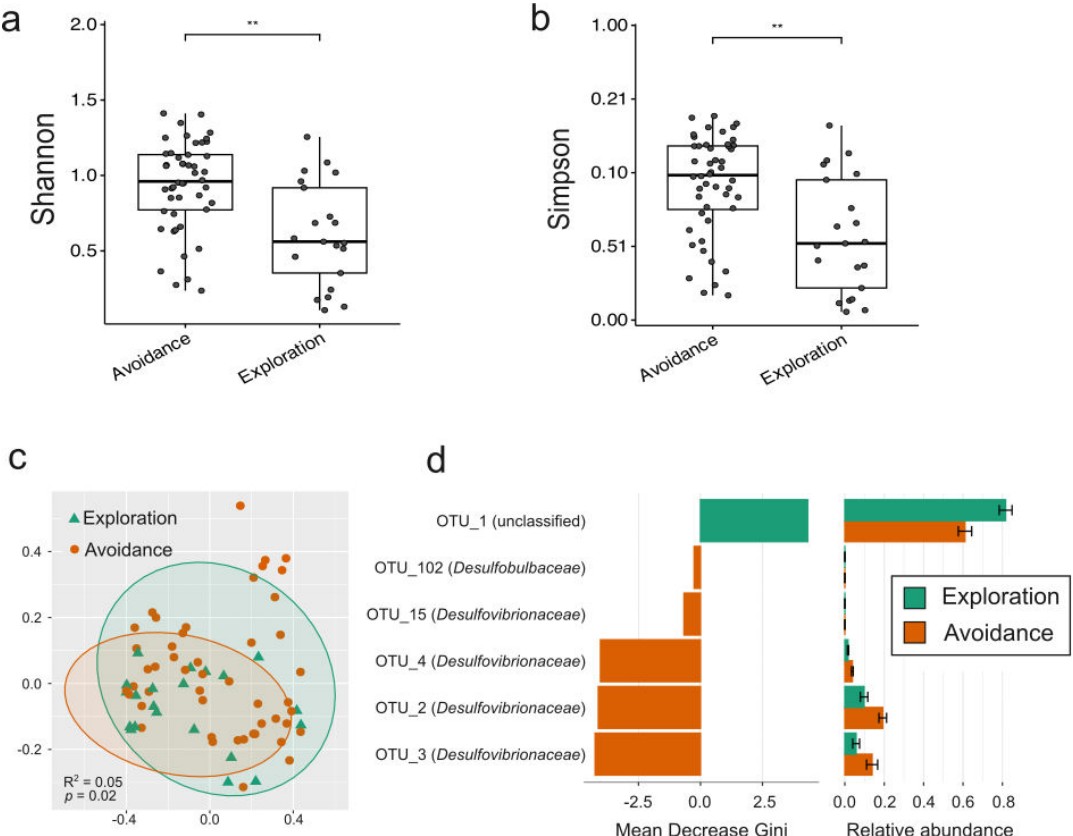

**FIG 5** Variation of sulfite-reducing bacteria among different personality groups. Alpha diversity (a and b) and beta diversity (c) showed significant differences between Avoidance and Exploration groups. (d) Indicators of SRB among Exploration and Avoidance personality groups. Indicator taxa were identified by random forest analysis. **$P < 0.01$, $t$-test. Mean Decrease Gini measures the extent to which the identified taxa contribute to the classification of personality groups.

associated with fecal sulfate, we were unable to obtain reliable measurements of $H_2S$ production from fecal samples due to strong background coloration interfering with the Cline assay (78). Future studies that integrate multi-omics approaches with functional readouts, including direct $H_2S$ quantification, metabolomics, and transcriptomics, will be essential to establish mechanistic links between SRB activity and behavioral phenotypes.

Previous studies have found conserved microbiota profiles among co-housed marmosets (13). Consistently, there were no significant differences in alpha diversity of the gut microbiota between our laboratory-bred marmosets that were housed in a controlled environment and received the same diet. Herein, we show that beta diversity of the gut microbiota was associated with marmoset personality as well as their social group, sex, age class, and breeding status. Group members shared more similar gut microbiota compared to individuals from different groups. Yet, relatedness did not increase gut microbiota similarity. This is consistent with previous findings, e.g., in zebrafish (*Danio rerio*) and Damaraland mole-rats (*Fukomys damarensis*), where environmental effects rather than host genetics or relatedness determine gut microbiota similarity (79, 80). Similarly, effects of the social environment on microbiota composition were found in chimpanzees and wild baboons when controlled for diet and genetic relatedness (22, 81).

Throughout the 2-year study period, consistent taxa were identified in different age classes, with *Helicobacter* enriched in juvenile and *Desulfovibrio* enriched in older adult marmosets (Fig. S4c and d). Consistent with previous aging studies in humans and other primates, a higher relative abundance of *Desulfovibrio* and *Bilophila* was observed in the older adult marmosets (82, 83). Similar enrichment of these taxa has been observed in

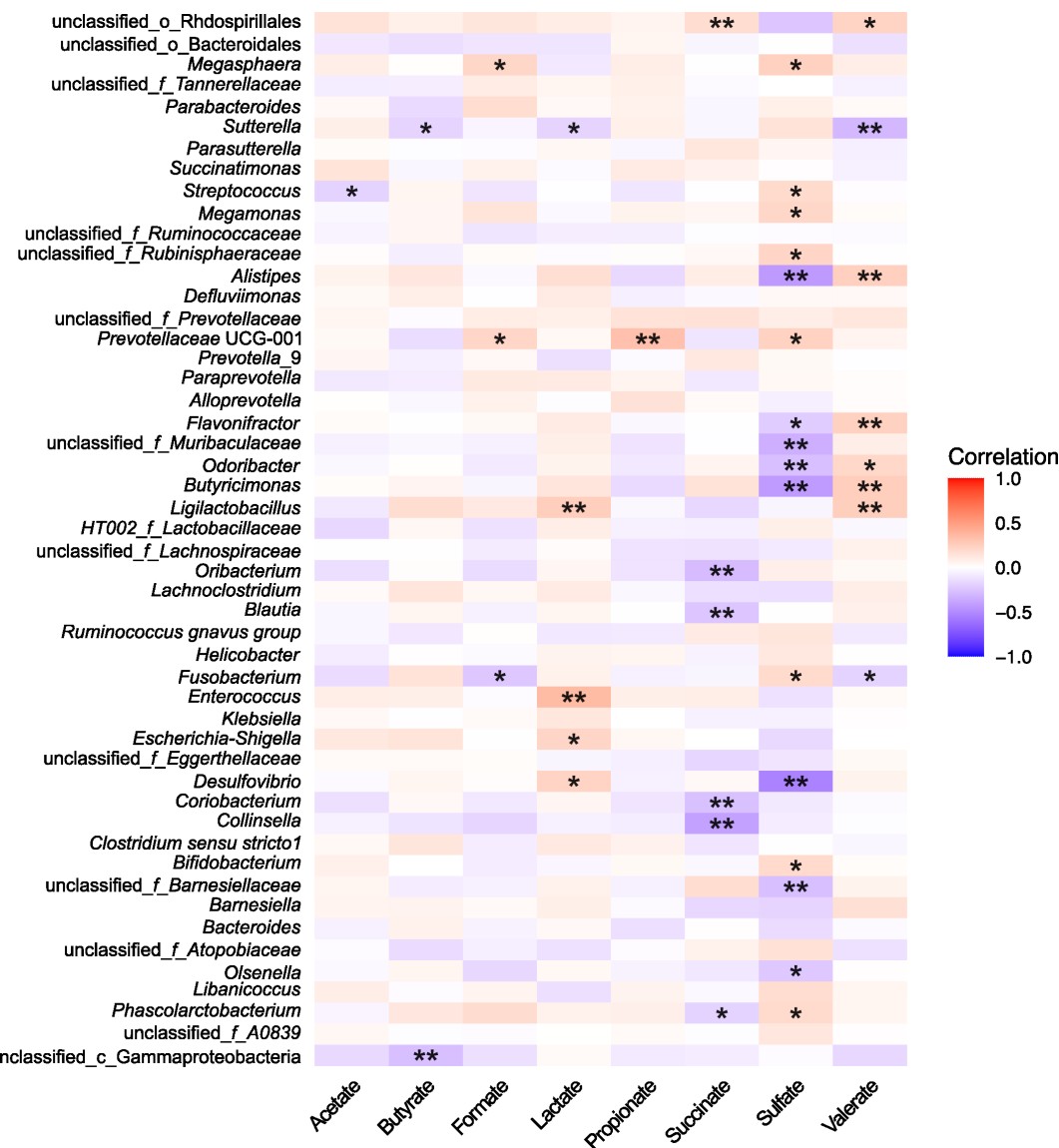

**FIG 6** Correlation between the gut microbiota (genus level) and short-chain fatty acids and sulfate in fecal samples collected in 2017 ($n$ = 123). The heatmap was constructed according to Spearman's correlation between the top 50 most abundant genera and fecal SCFAs. The degree of correlation is represented by color: red represents a positive correlation, and blue represents a negative correlation. The asterisks indicate the significance of the associations. *$P$ < 0.05 and **$P$ < 0.01.

aging mice (84). Additionally, aging in mice is linked to an increase in taurine-conjugated bile acids (85, 86). Notably, marmosets, like mice, exclusively conjugate bile acids with taurine (87). However, the effect of aging on taurine-conjugated bile acids in marmosets remains unclear. The enrichment of *Helicobacter* was previously reported in crab-eating infant macaques (82). While some *Helicobacter* species/strains are pathogenic, both pathogenic and non-pathogenic *Helicobacter* strains may be present in the intestinal tract (88, 89). For instance, *Helicobacter macacae* was enriched in rhesus macaque infants (<8-month-old) without diarrhea symptoms, and its abundance was reduced with the maturation of macaques (90). Thus, the enrichment of *Helicobacter* in juvenile marmosets may not be harmful, and its role remains to be elucidated.

Personality traits were also significantly linked with the beta diversity of gut microbiota in common marmosets. Specifically, *Megasphaera* was enriched in marmosets scoring higher on Exploration, whereas *Desulfovibrio* was enriched in more Avoidant-scoring marmosets (Fig. 3a and b). Similar to our results, in a study of older human adults,

*Megasphaera* was significantly associated with Extraversion, whereas *Desulfovibrio* was significantly associated with depression and Neuroticism (91). Additionally, the *dsrB* gene amplicon sequencing results confirmed that *Desulfovibrio* was enriched in the Avoidance group. Conversely, an uncultured SRB (*dsrB* OTU_1) was more abundant in the more explorative than in the more avoidant marmosets. Furthermore, *Bilophila* had a higher relative abundance in marmosets scoring high compared to marmosets scoring low on Stress/Activity. The relative abundance of *Desulfovibrio* is negatively correlated with the fecal sulfate concentration, suggesting the potential metabolism of sulfate by *Desulfovibrio*. UBA1819, a member of the *Ruminococcaceae* family, which is known for producing SCFAs and being closely related to anxiety-like behavior (92), also showed increased relative abundance in the marmosets showing higher Stress/Activity. These findings are consistent with previous studies in murine models, where an increase in *Desulfovibrionaceae* and *Ruminococcaceae* was observed under chronic or single prolonged stress (93–96). Yet, the role of those microbes under stress remains unknown. *Bacteroides* was consistently associated with the more avoidant marmosets, using differential analysis or Maaslin2 (Fig. 3b; Fig. S5). *Bacteroides* species enriched in the gut microbiome from major depressive disorder patients were shown to impact the susceptibility to depressive behavior. Notably, *Bacteroides fragilis*, *Bacteroides uniformis*, and *Bacteroides caccae* had a negative impact, while *Bacteroides ovatus* did not have a negative impact (97).

While our study provides novel insights into associations between gut microbiota and personality traits in captive common marmosets, several limitations should be acknowledged. First, the number of individuals included is modest, which inevitably limits statistical power and the generalization of our findings. Although we partially compensated for this by collecting repeated fecal samples per individual across 2 years, validation in larger and independent cohorts will be essential to confirm the associations reported here. Second, our cross-sectional design does not allow us to conclude directionality or causality; the observed associations should therefore be interpreted with caution. Longitudinal or interventional approaches will be needed to clarify whether microbiota influence behavioral phenotypes, whether behavior affects microbiota, or whether both are shaped by unmeasured factors. Third, our analyses were based on relative abundance data derived from 16S rRNA gene amplicon sequencing. Relative data are compositional and can obscure true changes in microbial load. Current state-of-the-art methods for quantitative microbiome profiling (e.g., spike-in standards or flow cytometry) (98) provide absolute abundance information and would strengthen inference in future studies. Finally, although we identified associations between specific taxa and personality traits, these links remain correlational; clarifying whether and how such taxa contribute to personality-related phenotypes will require mechanistic work—such as longitudinal designs, controlled interventions, and fecal microbiota transplantation experiments in appropriate animal models—coupled with functional readouts. Moreover, while the personality components assessed here map onto well-described behavioral syndromes, whether microbiome-personality associations translate into differences in ecological or social performance in natural settings remains a hypothesis that warrants testing in wild or semi-natural populations. Together, these limitations highlight that our findings are hypothesis-generating and require validation in larger and independent cohorts.

Despite these limitations, our study provides the first evidence for associations between personality traits and gut microbiota composition in a neotropical primate species under controlled dietary and housing conditions. We also show that social group membership and environmental factors explain more microbiota variation than genetic relatedness, underscoring the importance of social environment in shaping microbial communities.

## ACKNOWLEDGMENTS

The authors thank Alexandra Bohmann and the animal-keeping team for animal care, the staff of the Joint Microbiome Facility of the Medical University of Vienna and the

University of Vienna for performing amplicon sequencing, and Thomas Rattei and his team for maintaining and providing access to the Life Science Compute Cluster (LISC, University of Vienna).

This research was funded by the Austrian Science Fund (FWF) (grant DOI 10.55776/ I2320, 10.55776/DOC69, and 10.55776/COE7) and the China Scholarship Council (Ph.D. fellowship grant No. 201606850092 to H.Y.).

H.Y., V.Š., T.B., and A.L. conceived the study, with help from B.T.H. H.Y. collected the fecal samples and performed the microbial analysis. V.Š. performed the behavioral experiments and interpreted the data. H.Y. and V.Š. wrote the article, with help from A.L., C.W.H., P.P., and J.S., and B.H. contributed to the bioinformatic analyses. All authors read, revised, and approved the manuscript.

## AUTHOR AFFILIATIONS

[1]Division of Microbial Ecology, Centre for Microbiology and Environmental Systems Science, University of Vienna, Vienna, Austria

[2]Doctoral School in Microbiology and Environmental Science, Centre for Microbiology and Environmental Systems Science, University of Vienna, Vienna, Austria

[3]APC Microbiome Ireland, University College Cork, Cork, Ireland

[4]Department of Ecology and Evolution, University of Lausanne, Lausanne, Switzerland

[5]The Sense Innovation and Research Center, Lausanne & Sion, Lausanne, Switzerland

[6]Department of Behavioural and Cognitive Biology, University of Vienna, Vienna, Austria

[7]Bioscience Division, Los Alamos National Laboratory, Los Alamos, New Mexico, USA

[8]Joint Microbiome Facility of the Medical University of Vienna and the University of Vienna, Vienna, Austria

[9]Division of Clinical Microbiology, Department of Laboratory Medicine, Medical University of Vienna, Vienna, Austria

[10]Te Kura Pūtaiao Koiora, School of Biological Sciences, Te Whare Wānanga o Waitaha, University of Canterbury, Christchurch, New Zealand

## AUTHOR ORCIDs

Huimin Ye http://orcid.org/0000-0002-4706-9852
Vedrana Šlipogor http://orcid.org/0000-0002-8842-4144
Buck T. Hanson http://orcid.org/0000-0001-6947-7286
Joana Séneca http://orcid.org/0000-0003-3951-3674
Bela Hausmann http://orcid.org/0000-0002-0846-1202
Craig W. Herbold http://orcid.org/0000-0003-3479-0197
Petra Pjevac http://orcid.org/0000-0001-7344-302X
Thomas Bugnyar http://orcid.org/0000-0002-6072-9667
Alexander Loy http://orcid.org/0000-0001-8923-5882

## FUNDING

| Funder | Grant(s) | Author(s) |
| --- | --- | --- |
| Austrian Science Fund | 10.55776/I2320, 10.55776/DOC69, 10.55776/ COE7 | Alexander Loy |
| China Scholarship Council | 201606850092 | Huimin Ye |

## AUTHOR CONTRIBUTIONS

Huimin Ye, Conceptualization, Data curation, Formal analysis, Investigation, Methodology, Project administration, Validation, Visualization, Writing – original draft, Writing – review and editing | Vedrana Šlipogor, Conceptualization, Data curation, Formal analysis,

Investigation, Methodology, Project administration, Validation, Writing – original draft, Writing – review and editing | Buck T. Hanson, Methodology, Supervision, Visualization, Writing – review and editing | Joana Séneca, Data curation, Formal analysis, Writing – review and editing | Bela Hausmann, Data curation, Formal analysis, Writing – review and editing | Craig W. Herbold, Methodology, Validation, Writing – review and editing | Petra Pjevac, Data curation, Formal analysis, Methodology, Writing – review and editing | Thomas Bugnyar, Conceptualization, Supervision, Writing – review and editing | Alexander Loy, Conceptualization, Funding acquisition, Investigation, Methodology, Supervision, Writing – review and editing

## DATA AVAILABILITY

All 16S rRNA and dsrB gene sequence data presented in this article are available in the NCBI repository under BioProject accession number PRJNA1161472. The RScript used for the analyses of the data is available at https://doi.org/10.5281/zenodo.14860567.

## ETHICS APPROVAL

The research on marmoset personality was approved by the Animal Ethics and Experimentation Board, Faculty of Life Sciences, University of Vienna (Approval Number 2015-13), and adhered to the legal requirements of Austria. The study also adhered to the American Society of Primatologists' principles for the ethical treatment of primates. The housing conditions were in accordance with Austrian legislation and the European Association of Zoos and Aquaria (EAZA).

## ADDITIONAL FILES

The following material is available online.

### Supplemental Material

**Supplemental figures (Spectrum00443-25-s0001.docx).** Figures S1 to S5.
**Supplemental tables (Spectrum00443-25-s0002.xlsx).** Tables S1 to S9.

### Open Peer Review

**PEER REVIEW HISTORY (review-history.pdf).** An accounting of the reviewer comments and feedback.

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
