## [Reviewer comments · Microbiology Spectrum]

Microbiology Spectrum

Associations between gut microbiota and personality traits: insights from a captive common marmoset (*Callithrix jacchus*) colony

Huimin Ye, Vedrana Šlipogor, Buck Hanson, Joana S neca Cardoso da Silva, Bela Hausmann, Craig Herbold, Petra Pjevac, Thomas Bugnyar, and Alexander Loy

Corresponding Author(s): Huimin Ye, Universitat Wien

Review Timeline:

Submission Date:	February 15, 2025
Editorial Decision:	July 22, 2025
Revision Received:	September 15, 2025
Accepted:	October 15, 2025

Editor: Qi Su

Reviewer(s): Disclosure of reviewer identity is with reference to reviewer comments included in decision letter(s). The following individuals involved in review of your submission have agreed to reveal their identity: Aria Hajihassan (Reviewer #1)

Transaction Report:

DOI: <https://doi.org/10.1128/spectrum.00443-25>

Re: Spectrum00443-25 (**Associations between gut microbiota and personality traits: insights from a captive common marmoset (*Callithrix jacchus*) colony**)

Dear Dr. Huimin Ye:

Thank you for the privilege of reviewing your work. Below you will find my comments, instructions from the Spectrum editorial office, and the reviewer comments.

Revision Guidelines

Sincerely,
Qi Su
Editor
Microbiology Spectrum

Dear Dr. Huimin Ye:

Thank you for the privilege of reviewing your work. Below you will find my comments, instructions from the Spectrum editorial office, and the reviewer comments.

Revision Guidelines

Sincerely,
Qi Su
Editor
Microbiology Spectrum

>Response

Dear Dr. Su,

Thank you very much for your decision letter and for the opportunity to revise our manuscript. We sincerely appreciate the constructive feedback provided by you and the reviewer. We have carefully addressed all comments and revised the manuscript accordingly.

In particular, we have revised the wording throughout the manuscript to avoid implying causality, and we added a paragraph in the Discussion explicitly acknowledging the limitations of our study design. We believe these changes strengthen the manuscript and better position our findings in the context of current research.

We hope that the revised version meets the requirements of *Microbiology Spectrum*. Thank you again for considering our work.

1. Causality is Not Addressed

The manuscript reports associations between gut microbiota and personality traits but does not attempt to disentangle correlation from causation. While the authors acknowledge this in passing, more discussion is needed on the limitations of the study design in establishing causal relationships.

>Response

We thank the reviewer for this important comment. We agree that our study design does not allow for causal inference and that associations between microbiota composition and personality traits should be interpreted with caution. In response, we have carefully revised the manuscript to ensure that our wording does not imply

causality. Furthermore, we have expanded the discussion section to explicitly address the limitations of our study in this regard. Specifically, we now emphasize that our cross-sectional design precludes conclusions about causality (LINE 599-601). We also highlight that our study relied on relative abundance data, which has inherent limitations compared to quantitative microbiome profiling approaches¹ (LINE 604-608). This additional discussion better frames our findings as associations that warrant further validation in longitudinal or interventional studies.

2. Lack of Behavioral Contextualization

The behavioral trait definitions (e.g., "Exploration-Avoidance", "Boldness-Shyness") are largely based on PCA without sufficient clarification of how these map onto ethologically relevant behaviors. The link between microbial patterns and actual ecological or social behaviors remains speculative.

>Response

We thank the reviewer for this helpful suggestion. The personality components used here are derived from an established, repeatable test battery previously validated in the same marmoset colony². That study demonstrated temporal and contextual consistency of behavioral variables across five standardized tests (General Activity, Novel Object, Novel Food, Foraging Under Risk, Predator), and extracted three components with clear ethological meaning:

Exploration–Avoidance: tendencies to approach, inspect, and manipulate novel stimuli (neophilia, object/food exploration, interaction latencies).

Boldness–Shyness: response to risk/threat, proximity to stimuli, vigilance/withdrawal in predator-related and risk contexts.

Stress/Activity: locomotor activity and stress-related behaviors (e.g., movement, agitation) across contexts.

To address the reviewer's request for clearer mapping, we have revised the Methods to link each axis to ethologically relevant tendencies (neophilia/exploration, risk-taking/avoidance, and stress-related activity) (LINE 288-293). Finally, we acknowledge in the Discussion that while these components map onto well-described behavioral syndromes, connecting microbiome–personality associations to ecological performance in the wild remains a hypothesis that warrants testing in naturalistic settings (LINE 612-616).

3. Limited Sample Size

The sample size of 26 individuals is relatively small, particularly when split into subgroups for personality traits, sex, age class, and breeding status. This compromises statistical power and increases the risk of false-positive findings, especially in multi-comparison settings.

>Response

We acknowledge the reviewer's concern regarding the relatively small number of individuals. This is an inherent limitation of studies on non-human primates, where ethical and logistical constraints often preclude large sample sizes. To partially address this limitation, we collected repeated fecal samples from each individual across two sampling periods, which increased the robustness of our within- and between-individual comparisons.

In the revised Discussion, we have explicitly acknowledged that the modest number of individuals limits statistical power and generalizability, and that our results should be interpreted as hypothesis-generating. We emphasize that validation in larger and independent cohorts will be essential to confirm the associations reported here (LINE 597-599).

4. Overreliance on Machine Learning (Random Forest)

The use of Random Forests to identify differential taxa lacks transparency in feature selection and variable importance thresholds. There's a risk of overfitting given the small dataset, and the lack of validation or cross-validation procedures weakens the robustness of the identified indicator taxa.

>Response

We thank the reviewer for raising this important point. We would like to clarify that Random Forest was not used as the sole or primary method for testing associations between taxa and personality traits. Rather, we applied Random Forest classification in an exploratory manner to highlight taxa that discriminated between personality-based groups. For the inferential analyses, we relied on MaAsLin2, which models associations between continuous personality scores and microbial taxa while adjusting for covariates (sex, age class, breeding status, and social group) and correcting for multiple testing.

We also explicitly state in the Discussion that given the relatively small sample size, Random Forest results should be considered illustrative and hypothesis-generating rather than confirmatory (LINE 608-612). We have now added that taxa with MeanDecreaseGini > 0 were retained as differential taxa in Methods (LINE 298-300).

5. Weak Justification for SRB Focus

The emphasis on sulfite-reducing bacteria (SRB) is introduced without a strong rationale. The mechanistic link between SRB abundance and behavior is poorly developed and seems somewhat opportunistic. There is no functional validation of the microbial metabolism (e.g., H₂S production assays or host physiological readouts).

>Response

We thank the reviewer for this valuable comment. Our rationale for focusing on sulfite-reducing bacteria (SRB) is that they constitute a distinct functional guild in the gut microbiome, producing hydrogen sulfide (H₂S) via dissimilatory sulfite reduction. H₂S is a biologically active metabolite with both protective and deleterious effects on host physiology and cognition, and prior work in rodents and humans has linked SRB, particularly *Desulfovibrio*, to behavioral phenotypes, stress responses, and age-related conditions. Given these reported associations, and our observation that *Desulfovibrio* was enriched in more avoidant individuals, we considered it important to examine SRB more closely in this study using targeted *dsrB* amplicon sequencing (LINE 111-120).

We agree that our study does not include direct functional validation of SRB activity. We attempted H₂S quantification but were unable to obtain reliable results, and unfortunately no fixed material remains available to repeat these assays. We have now clarified this in the revised manuscript and emphasized in the Discussion that the absence of functional readouts (e.g., H₂S production assays, metabolomics, transcriptomics) represents an important limitation. We also explicitly note that our SRB findings are exploratory and hypothesis-generating, and that future work combining multi-omics and functional assays will be required to establish mechanistic links between SRB activity and behavioral phenotypes (LINE 537-543).

1. Vandeputte, D. *et al.* Quantitative microbiome profiling links gut community variation to microbial load. *Nature* **551**, 507–511 (2017).
2. Šlipogor, V., Graf, C., Massen, J. J. M. & Bugnyar, T. Personality and social environment predict cognitive performance in common marmosets (*Callithrix jacchus*). *Sci. Rep.* **12**, 6702 (2022).

Re: Spectrum00443-25R1 (**Associations between gut microbiota and personality traits: insights from a captive common marmoset (*Callithrix jacchus*) colony**)

Dear Dr. Huimin Ye:

Your manuscript has been accepted, and I am forwarding it to the ASM production staff for publication. Your paper will first be checked to make sure all elements meet the technical requirements. ASM staff will contact you if anything needs to be revised before copyediting and production can begin. Otherwise, you will be notified when your proofs are ready to be viewed.

Sincerely,
Qi Su
Editor
Microbiology Spectrum